# Fractons on curved spacetime in 2 + 1 dimensions

Jelle Hartong[1⋆], Giandomenico Palumbo[2†], Simon Pekar[3‡],
Alfredo Perez[4,5∘] and Stefan Prohazka[6§]

**1** School of Mathematics and Maxwell Institute for Mathematical Sciences,
University of Edinburgh, Peter Guthrie Tait road, Edinburgh EH9 3FD, UK
**2** School of Theoretical Physics, Dublin Institute for Advanced Studies,
10 Burlington Road, Dublin 4, Ireland
**3** Centre de Physique Théorique - CPHT, École polytechnique,
CNRS, Institut Polytechnique de Paris,
91120 Palaiseau Cedex, France
**4** Centro de Estudios Científicos (CECs), Avenida Arturo Prat 514, Valdivia, Chile
**5** Facultad de Ingeniería, Arquitectura y Diseño, Universidad San Sebastián,
sede Valdivia, General Lagos 1163, Valdivia 5110693, Chile
**6** University of Vienna, Faculty of Physics, Mathematical Physics,
Boltzmanngasse 5, 1090, Vienna, Austria

⋆ j.hartong@ed.ac.uk , † giandomenico.palumbo@gmail.com ,
‡ simon.pekar@polytechnique.edu , ∘ alfredo.perez@uss.cl , § stefan.prohazka@univie.ac.at

## Abstract

We study dipole Chern–Simons theory with and without a cosmological constant in 2 + 1 dimensions. We write the theory in a second order formulation and show that this leads to a fracton gauge theory coupled to Aristotelian geometry which can also be coupled to matter. This coupling exhibits the remarkable property of generalizing dipole gauge invariance to curved spacetimes, without placing any limitations on the possible geometries. We also use the second order formulation to construct a higher dimensional generalization of the action. Finally, for the (2 + 1)-dimensional Chern–Simons theory we find solutions and interpret these as electric monopoles, analyze their charges and argue that the asymptotic symmetries are infinite-dimensional.

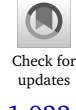

# 1   Introduction

Fractons [1, 2] are novel, at this point theoretical, quasiparticles with the distinctive feature of having only limited mobility [3–5]. Their underlying (exotic) dipole symmetry falls into the broader class of generalized symmetries [6, 7] that challenge, and hence improve, our understanding of quantum field theories.

One puzzling aspect is their coupling to spacetime [8, 9]. While the matter fields [10] can be coupled to generic Aristotelian geometry [11, 12] the gauge theory [13, 14] that mediates the forces puts restrictions on the admissible spacetimes [9, 11, 12]. The reason for the restriction lies in the tension between general Aristotelian covariance and dipole gauge symmetries. In Cartesian coordinates the latter gauge transformations with parameter $\Lambda$ act on the gauge fields $\phi$ and symmetric tensor $A_{ij}$ as

$$\delta_\Lambda \phi = \partial_t \Lambda, \qquad \delta_\Lambda A_{ij} = -\partial_i \partial_j \Lambda, \tag{1}$$

where $i, j$ are spatial indices. They couple to matter via $\rho\,\delta\phi + J^{ij}\delta A_{ij}$, which leads to the conservation equation

$$\partial_t \rho + \partial_i \partial_j J^{ij} = 0, \tag{2}$$

which is at the heart of many of the interesting properties of this theory. It implies for example the conservation of the electric charge $Q = \int \rho\, dx$ and the dipole moment $\vec{D} = \int \vec{x}\rho\, dx$ and that isolated monopoles cannot move. These relations also show that this theory is non-lorentzian and that generalizing it to generic curved spacetimes is nontrivial.

In this work we will show that it is possible to couple fracton gauge fields consistently to a particular Aristotelian theory of gravity.[1] We circumvent the earlier no-go results by providing another gauge theory, which derives from gauging the fracton/dipole algebra in a spirit similar to the gauging of spacetime symmetry algebras to obtain Einstein gravity in the first order formulation (see, e.g., [21, 22]).[2] In $2 + 1$ dimensions this leads to a fracton/dipole Chern–Simons (CS) theory [26].[3] One of our main results is to translate this theory into second order formulation (see Section 3 for the definition of all expressions)

$$S[\phi, A_{\mu\nu}, \tau_\mu, h_{\mu\nu}] = \int d^3x e\Big( -\mu\phi h^{\mu\nu}R_{\mu\nu} + 2\mu K_{\mu\rho}A_{\nu\sigma}(h^{\mu\nu}h^{\rho\sigma} - h^{\mu\rho}h^{\nu\sigma}) \\ + \frac{\mu_H}{2}\varepsilon^{\rho\sigma\kappa}\tau_\kappa(\partial_\rho\tau_\sigma - \partial_\sigma\tau_\rho)\Big). \tag{3}$$

This action provides a coupling of fracton gauge fields $(\phi, A_{\mu\nu})$ to the Aristotelian geometry given by $(\tau_\mu, h_{\mu\nu})$ and can be generalized to generic spacetime dimension. Since the gauge fields act as Lagrange multipliers for the geometry, it exhibits similarities to JT gravity [31, 32] and BF models.

This coupling possesses the remarkable property that the action remains invariant under the following generalization of dipole gauge transformations to curved spacetimes, without requiring additional restrictions on the geometry

$$\delta\phi = n^\mu\partial_\mu\bar{\Lambda}, \qquad \delta A_{\mu\nu} = -P^\rho_{(\mu}P^\sigma_{\nu)}\nabla_\rho\partial_\sigma\bar{\Lambda}, \tag{4}$$

where $P^\rho_\mu = h_{\mu\nu}h^{\nu\rho} = \delta^\rho_\mu - n^\rho\tau_\mu$ is the spatial projector and $n^\mu$ is the vector dual to the clock form $\tau_\mu$, i.e., $n^\mu\tau_\mu = 1$ and $n^\mu h_{\mu\nu} = 0$. This implies a generalization of dipole conservation, i.e., $\partial_\mu(eJ^\mu) = 0$ where $e$ is the integration measure (analog of $\sqrt{-g}$ in a relativistic setup) and where $J^\mu$ is the current

$$J^\mu = \rho n^\mu + \nabla_\nu\left(P^\mu_\rho P^\nu_\sigma J^{\rho\sigma}\right), \tag{5}$$

where for simplicity we have assumed that the Aristotelian metric-compatible affine connection $\nabla_\mu$ has no torsion. These are the curved generalizations of (1) and (2) (without linearization or further restrictions on the geometry). Hence dipole gauge invariance puts no restrictions on the geometry (in any dimension).

Following [11, 12] we show how we can couple the $(2 + 1)$-dimensional theory to matter theories [10] (Section 3.5) and how we can add a cosmological constant term. In $2 + 1$ dimensions we also construct a solution to the nonlinear equations and derive for negative and vanishing cosmological constant the conserved charges. The geometry is spherically symmetric and has nonzero electric charge and energy and we therefore interpret it as a monopole. For negative $\Lambda$ we also discuss the asymptotic symmetries which are given by an infinite dimensional enhancement of the fracton algebra (cf., [33, 34]).

This work is structured as follows. In Section 2 we introduce the fracton Chern–Simons theory, i.e., the first order formulation, with and without cosmological constant term. In Section 3 we discuss the underlying Aristotelian geometry and translate to second order formulation, which we use to generalize the action to generic dimensions (Section 3.4). In $2 + 1$ dimensions we show that we can couple the theory to matter fields (Section 3.5). In Section 4 we find static circularly symmetric solutions, interpret them as monopoles and discuss their

---

[1] For complementary approaches, see, e.g., [15–20].

[2] The gauging of these symmetries and their relation to Aristotelian geometry has also been discussed in the context of hydrodynamics [23–25].

[3] The existence of this CS theory already follows from the correspondence of the fracton and Carroll symmetries [11, 27, 28] and the fact that theories with Carroll symmetry allow for a CS formulation [29, 30].

charges and asymptotic symmetries. We close by mentioning various interesting generalizations (Section 5). We have delegated technical aspects concerning the Aristotelian connection to Appendix A, its curvature to Appendix B and Lie algebraic considerations to Appendix C.

## 2 Fracton Chern-Simons theory in $2 + 1$ spacetime dimensions

In this section we introduce a Chern-Simons theory based on the fracton algebra, with and without cosmological constant.

### 2.1 Fracton algebra and its invariant metric

The fracton/dipole algebra [35] in $2 + 1$ dimensions is spanned by the set of generators $\langle J, H, P_a, Q, D_a \rangle$, which are the usual generators of symmetry of Aristotelian spacetime, i.e. spatial rotations, time and space translations, dual to the angular momentum, energy and linear momentum, as well as two generators of internal symmetry dual to electric and dipole charge, respectively. The non-vanishing commutation relations are given by

$$[J, P_a] = \epsilon_{ab} P_b, \qquad\qquad [J, D_a] = \epsilon_{ab} D_b, \qquad\qquad [P_a, D_b] = \delta_{ab} Q, \qquad (6)$$

where $a, b = 1, 2$ and $\epsilon_{12} = 1$. It is a nonsemisimple algebra with a nontrivial central extension ($Q$) and a trivial one ($H$). In $2 + 1$ dimensions there exist other nontrivial central extensions, but since they do not persist for generic dimensions we will not consider them.

If we want to use the symmetries (6) to construct a Chern–Simons theory one usually requires the existence of an invariant metric, that is a symmetric, ad-invariant, non-degenerate bilinear form on the Lie algebra. The fact that this algebra is nonsemisimple makes the existence of such an invariant metric nontrivial. In contradistinction, for semisimple Lie algebras there is of course always the Killing form (by Cartan's criterion). For the case at hand, the existence follows from the isomorphism of the Carroll and fracton/dipole algebras [11], and the fact that the Carroll algebra has an invariant metric in $2 + 1$ dimensions [29, 30].

For the fracton algebra (6) the most general invariant metric is given by

$$\langle J, Q \rangle = \mu, \qquad\qquad \langle P_a, D_b \rangle = -\mu \epsilon_{ab}, \qquad\qquad \langle H, H \rangle = \mu_H,$$
$$\langle J, J \rangle = \chi_J, \qquad\qquad \langle J, H \rangle = \chi_{JH}, \qquad\qquad\qquad\qquad\qquad (7)$$

which is non-degenerate for $\mu \neq 0 \neq \mu_H$. We will see below that since $\mu_H \neq 0$, one can without loss of generality always set $\chi_{JH}$ equal to zero in the CS action.

### 2.2 Fracton/dipole CS action

With these ingredients we can write a Chern–Simons theory

$$S_{\mathrm{CS}}[A] = \int \langle A \wedge dA + \tfrac{1}{3}[A, A] \wedge A \rangle \equiv \int L_{\mathrm{CS}}, \qquad (8)$$

for the Lie algebra valued one-form $A$, decomposed as

$$A = A_t dt + A_i dx^i = \tau H + e^a P_a + \omega J + aQ + A^a D_a. \qquad (9)$$

The Chern–Simons action for the fracton algebra (6) with invariant metric (7) is given by

$$S[\tau, e, \omega, a, A] = \int 2\mu \left( \omega \wedge da - \epsilon_{ab} e^a \wedge dA^b + e^a \wedge A^a \wedge \omega \right) + \mu_H \tau \wedge d\tau$$
$$+ \chi_J \omega \wedge d\omega + 2\chi_{JH} \omega \wedge d\tau. \qquad (10)$$

This theory was already discussed in [26], where the term proportional to $\mu_H$ was mentioned, but left implicit. Since $\tau$ is a relevant part of the Aristotelian geometry we will keep it explicit.

Using the fact that $\mu_H$ was assumed to be nonzero, we can rewrite the last three terms of the above action as

$$\mu_H \tau \wedge d\tau + \chi_J \omega \wedge d\omega + 2\chi_{JH} \omega \wedge d\tau = \mu_H (\tau + \alpha\omega) \wedge d(\tau + \alpha\omega) + \beta \omega \wedge d\omega \tag{11}$$

(up to a total derivative), where $\alpha$ and $\beta$ are given by

$$\alpha = \chi_{JH}/\mu_H, \qquad \beta = \chi_J - \chi_{JH}^2/\mu_H. \tag{12}$$

By performing a redefinition of $\tau$, given by $\tau' = \tau + \alpha\omega$, we can remove the term with $\chi_{JH}$ entirely, and so without loss of generality we can set

$$\chi_{JH} = 0. \tag{13}$$

For simplicity we will furthermore assume that

$$\chi_J = 0. \tag{14}$$

Like every CS theory in $2 + 1$ dimensions, this theory has no local propagating degrees of freedom and, without further input, it does not depend on any (non)lorentzian metric or geometry. In the next sections we will however interpret some of these generators in terms of Aristotelian geometry [11,12], e.g., we will impose additional restrictions on the vielbeine (see (24)). This means we introduce additional structure, which does not change the degrees of freedom, but the theory depends then on geometric quantities, like the Aristotelian analog of a metric.

## 2.3 Equations of motion, and gauge transformations

The equations of motion are given by the usual curvature equals zero equations where the curvature is

$$F = dA + \tfrac{1}{2}[A,A] = 0. \tag{15}$$

In components, if we vary the fields in the action, this amounts to

$$\delta\tau: \qquad d\tau = 0, \tag{16a}$$

$$\delta e^a: \qquad dA^a - \epsilon_{ab}\, \omega \wedge A^b = 0, \tag{16b}$$

$$\delta a: \qquad d\omega = 0, \tag{16c}$$

$$\delta\omega: \qquad da + e^a \wedge A^a = 0, \tag{16d}$$

$$\delta A^a: \qquad de^a - \epsilon_{ab}\, \omega \wedge e^b = 0. \tag{16e}$$

The gauge transformations are of the form $\delta A = d\varepsilon + [A, \varepsilon]$ where

$$\varepsilon = \lambda J + \zeta H + \zeta^a P_a + \Lambda Q + \Lambda^a D_a. \tag{17}$$

In components this reads

$$\delta\tau = d\zeta, \tag{18a}$$

$$\delta e^a = d\zeta^a + \lambda\, \epsilon^a{}_b\, e^b - \omega\, \epsilon^a{}_b\, \zeta^b, \tag{18b}$$

$$\delta\omega = d\lambda, \tag{18c}$$

$$\delta a = d\Lambda + e^a\, \Lambda_a - A^a\, \zeta_a, \tag{18d}$$

$$\delta A^a = d\Lambda^a + \lambda\, \epsilon^a{}_b\, A^b - \omega\, \epsilon^a{}_b\, \Lambda^b. \tag{18e}$$

In Section 3 we provide a more detailed and general analysis, but let us first give some intuition on how we can recover the dipole conservation (2) from the coupling to the fields $a$ and $A^a$ of the CS theory. We restrict ourselves to a flat background in Cartesian coordinates, i.e., $e_\mu{}^a = \delta_\mu^a$, $\tau_\mu = \delta_\mu^t$ and $\omega = 0$, which allows to simply replace tangent indices $a$, $b$ into spatial ones $i$, $j$. The charge and dipole gauge transformations are then given by

$$\delta a_i = \partial_i \Lambda + \Lambda_i\,, \qquad \delta a_t = \partial_t \Lambda\,, \qquad \delta A_i{}^j = \partial_i \Lambda^j\,, \qquad \delta A_t{}^j\,, = \partial_t \Lambda^j\,, \qquad (19)$$

and when we set $\Lambda_i = -\partial_i \Lambda$ (which is the residual gauge transformation of the gauge choice $a_i = 0$) we find that $a_t$ and $A_{ij}$ transform precisely like the gauge fields in (1), which implies the dipole conservation law.

## 2.4 Adding a cosmological constant

We can deform the fracton algebra (6) by adding curvature ("cosmological constant") $\Lambda$, which results in the algebra (see Appendix C for the details)

$$\begin{aligned}
[J, P_a] &= \epsilon_{ab} P_b\,, & [J, D_a] &= \epsilon_{ab} D_b\,, & [P_a, D_b] &= \delta_{ab} Q\,, \\
[Q, P_a] &= \Lambda D_a\,, & [P_a, P_b] &= \Lambda \epsilon_{ab} J\,, &&
\end{aligned} \qquad (20)$$

with the most general invariant metric

$$\begin{aligned}
\langle J, Q \rangle &= \mu\,, & \langle P_a, D_b \rangle &= -\mu\,\epsilon_{ab}\,, & \langle H, H \rangle &= \mu_H\,, \\
\langle J, J \rangle &= \chi_J\,, & \langle P_a, P_b \rangle &= \Lambda \chi_J\,\delta_{ab}\,, &&
\end{aligned} \qquad (21)$$

which is non-degenerate for $\mu \neq 0 \neq \mu_H$. With the connection (9) the CS action (10) is then given by

$$S_\Lambda[\tau, e, \omega, a, A] = \int 2\mu \left( \omega \wedge da - \epsilon_{ab} e^a \wedge dA^b + e^a \wedge A^a \wedge \omega + \frac{\Lambda}{2} \epsilon_{ab} e^a \wedge e^b \wedge a \right)$$
$$+ \mu_H \tau \wedge d\tau + \chi_J \left( \omega \wedge d\omega + \Lambda(e^a \wedge de^a + \epsilon_{ab} e^a \wedge e^b \wedge \omega) \right)\,, \qquad (22)$$

with equations of motions (for $\chi_J = 0$)

$$\delta\tau : \qquad\qquad\qquad d\tau = 0\,, \qquad (23a)$$

$$\delta e^a : \qquad dA^a - \epsilon_{ab} \omega \wedge A^b + \Lambda a \wedge e^a = 0\,, \qquad (23b)$$

$$\delta a : \qquad\qquad d\omega + \frac{1}{2}\Lambda \epsilon_{ab} e^a \wedge e^b = 0\,, \qquad (23c)$$

$$\delta\omega : \qquad\qquad\qquad da + e^a \wedge A^a = 0\,, \qquad (23d)$$

$$\delta A^a : \qquad\qquad de^a - \epsilon_{ab} \omega \wedge e^b = 0\,. \qquad (23e)$$

Equation (23c) shows that $\Lambda$ can be interpreted as adding a cosmological constant to the geometry.

# 3 Second-order formulation

In this section we will translate the Chern–Simons action to the second-order formulation. Roughly speaking and similar to general relativity, we integrate out $\omega$ and find a connection that is built out of the Aristotelian metric like fields $\tau_\mu$ and $h_{\mu\nu}$. We show that the resulting action is gauge invariant and derive how to couple it to matter. With the exception of the matter coupling we show how we can generalize to generic dimension.

### 3.1 Integrating out fields

We will assume that $(\tau_\mu, e_\mu{}^a)$ forms an invertible set of vielbeine whose inverse is given by $(n^\mu, e^\mu{}_a)$ where

$$n^\mu \tau_\mu = 1, \qquad e^a_\mu e^\nu_a + \tau_\mu n^\nu = \delta^\nu_\mu. \tag{24}$$

We define $\xi^\mu$ as $\zeta = \xi^\mu \tau_\mu$ and $\zeta^a = \xi^\mu e_\mu^a$, so that we have a bijective correspondence between $\xi^\mu$ and $(\zeta, \zeta^a)$. Using the equations of motion, i.e., that $F = 0$ it then follows that the $(\zeta, \zeta^a)$ transformations are on shell equivalent to Lie derivatives along $\xi^\mu$.

Since $(\tau_\mu, e_\mu{}^a)$ are invertible, the gauge transformation can also be written as

$$\delta A_\mu = \partial_\mu \varepsilon + \left[A_\mu, \varepsilon\right] = \mathcal{L}_\xi A_\mu + \partial_\mu \Sigma + \left[A_\mu, \Sigma\right] + \xi^\nu F_{\mu\nu} = \bar{\delta} A_\mu + \xi^\nu F_{\mu\nu}, \tag{25}$$

where the last equality defines $\bar{\delta} A_\mu$ and where

$$\varepsilon = \xi^\mu A_\mu + \Sigma, \qquad\qquad \Sigma = \bar{\lambda} J + \bar{\Lambda} Q + \bar{\Lambda}^a D_a. \tag{26}$$

Because the difference between $\delta A_\mu$ and $\bar{\delta} A_\mu$ is proportional to the equations of motion it follows that $\bar{\delta} A_\mu$ is also a gauge symmetry of the theory. Basically this is because we have $\xi^\sigma \langle F_{[\mu\nu} F_{\rho]\sigma} \rangle = 0$. The variation of the CS action is (up to boundary terms)

$$\delta S_{\mathrm{CS}} = 2 \int \langle F \wedge \delta A \rangle, \tag{27}$$

and the above conclusion follows from $\langle F \wedge i_\xi F \rangle = 0$ where $i_\xi$ denotes the interior product with respect to the vector $\xi$.

The equations of motion allow us to solve for some of the fields algebraically in terms of the other fields. If the set of fields we vary can be solved for that same set of fields algebraically we are allowed to substitute these back into the action and obtain an equivalent description in terms of fewer fields. Since the $\omega$ connection is one of these fields the resulting action will be a second order formulation of the theory.

Consider equations (16d) and (16e). The latter can be written as

$$\partial_\mu e_\nu{}^a - \partial_\nu e_\mu{}^a - \epsilon_{ab} \left(\omega_\mu e_\nu{}^b - \omega_\nu e_\mu{}^b\right) = 0. \tag{28}$$

By contracting this equation with $n^\mu$ and $e^\nu{}_c$ we can solve for $\omega_\mu$ leading to

$$n^\mu \omega_\mu = \frac{1}{2} \epsilon_{ac} n^\mu e^\nu{}_c \left(\partial_\mu e_\nu{}^a - \partial_\nu e_\mu{}^a\right), \tag{29a}$$

$$e^\mu{}_a \omega_\mu = \frac{1}{2} \epsilon_{cd} e^\mu_c e^\nu{}_d \left(\partial_\mu e_\nu{}^a - \partial_\nu e_\mu{}^a\right). \tag{29b}$$

Equation (16d) can be written as

$$\partial_\mu a_\nu - \partial_\nu a_\mu + e_\mu{}^a A_\nu{}^a - e_\nu{}^a A_\mu{}^a = 0, \tag{30}$$

from which it follows that

$$n^\mu A_\mu{}^a = n^\mu e^\nu{}_a \left(\partial_\mu a_\nu - \partial_\nu a_\mu\right), \tag{31a}$$

$$e^\mu{}_b A_\mu{}^a - e^\mu{}_a A_\mu{}^b = -e^\mu{}_a e^\nu{}_b \left(\partial_\mu a_\nu - \partial_\nu a_\mu\right). \tag{31b}$$

The second equation tells us that the most general solution to $A_\mu^a$ is given by

$$A_\mu{}^a = \frac{1}{2} e^\nu{}_a (\delta^\rho_\mu + n^\rho \tau_\mu)\left(\partial_\rho a_\nu - \partial_\nu a_\rho\right) + S^a{}_b e_\mu{}^b =: \tilde{A}_\mu{}^a + S^a{}_b e_\mu{}^b, \tag{32}$$

where $S^{ab}$ is symmetric in $a$ and $b$, but otherwise arbitrary, and the $a, b, \ldots$ indices are raised and lowered with a Kronecker delta, and where we defined $\tilde{A}_\mu{}^a$. The solution for $\omega$ is equivalent to imposing (16e) whereas the solution for $A^a = \tilde{A}^a + S^a{}_b e^b$ where $\tilde{A}^a$ obeys (16d). Using the solutions for $\omega$ and $A^a$ the Lagrangian can be rewritten (up to a total derivative) as

$$
\begin{aligned}
L_{\mathrm{CS}} &= 2\mu \left( \omega \wedge (da + e^a \wedge A^a) - \epsilon_{ab} de^a \wedge A^b \right) + \mu_H \tau \wedge d\tau \\
&= 2\mu \left( -\epsilon_{ab} de^a \wedge S^b{}_c e^c - \epsilon_{ab} de^a \wedge \tilde{A}^b \right) + \mu_H \tau \wedge d\tau \\
&= 2\mu \left( -\epsilon_{ab} de^a \wedge S^b{}_c e^c - \omega \wedge e^b \wedge \tilde{A}^b \right) + \mu_H \tau \wedge d\tau \\
&= 2\mu \left( -\epsilon_{ab} de^a \wedge S^b{}_c e^c + \omega \wedge da \right) + \mu_H \tau \wedge d\tau \\
&= 2\mu \left( a \wedge d\omega - \epsilon_{ab} de^a \wedge S^b{}_c e^c \right) + \mu_H \tau \wedge d\tau \,,
\end{aligned}
\tag{33}
$$

where $\omega$ is no longer an independent connection, but where $S^{ab} = S^{(ab)}$ is an independent variable (as are $\tau_\mu, e_\mu{}^a$ and $a_\mu$).

## 3.2 Aristotelian geometry

The goal is to rewrite (33) in terms of an affine connection and its associated curvature as well as possibly torsion terms of said affine connection. In order to introduce such a connection we invoke the following vielbein postulate

$$
0 = \partial_\mu \tau_\nu - \Gamma^\rho_{\mu\nu} \tau_\rho \,,
\tag{34a}
$$

$$
0 = \partial_\mu e_\nu{}^a - \epsilon^{ab} \omega_\mu e_{\nu b} - \Gamma^\rho_{\mu\nu} e_\rho{}^a \,.
\tag{34b}
$$

If we solve these two equations for $\Gamma^\rho_{\mu\nu}$ we obtain

$$
\Gamma^\rho_{\mu\nu} = n^\rho \partial_\mu \tau_\nu + e^\rho{}_b \left( \partial_\mu e_\nu{}^b - \epsilon^{bc} \omega_\mu e_{\nu c} \right) \,.
\tag{35}
$$

It can be shown (see appendix A) that for $\omega_\mu$ given in (29a) and (29b) we can write the affine connection as

$$
\Gamma^\rho_{\mu\nu} = n^\rho \partial_\mu \tau_\nu + \frac{1}{2} h^{\rho\sigma} \left( \partial_\mu h_{\nu\sigma} + \partial_\nu h_{\mu\sigma} - \partial_\sigma h_{\mu\nu} \right) - h^{\rho\sigma} \tau_\nu K_{\mu\sigma} \,,
\tag{36}
$$

where we defined

$$
h_{\mu\nu} = \delta_{ab} e_\mu{}^a e_\nu{}^b \,, \qquad h^{\mu\nu} = \delta^{ab} e^\mu{}_a e^\nu{}_b \,,
\tag{37}
$$

as well as

$$
K_{\mu\nu} = \frac{1}{2} \mathcal{L}_n h_{\mu\nu} \,.
\tag{38}
$$

This connection is metric compatible in the sense that

$$
\nabla_\mu \tau_\nu = 0 \,, \qquad \nabla_\mu h_{\nu\rho} = 0 \,,
\tag{39}
$$

which follows from the vielbein postulates and is thus true by design. This also implies that $\nabla_\mu n^\nu = \nabla_\mu h^{\nu\rho} = 0$. Furthermore, it has nonzero torsion. Explicitly the torsion is given by

$$
T^\rho_{\mu\nu} = 2\Gamma^\rho_{[\mu\nu]} = n^\rho \left( \partial_\mu \tau_\nu - \partial_\nu \tau_\mu \right) + h^{\rho\sigma} \left( \tau_\mu K_{\nu\sigma} - \tau_\nu K_{\mu\sigma} \right) \,.
\tag{40}
$$

The torsion is thus determined by $d\tau$ and $K_{\mu\nu}$. One could formulate this as saying that the torsion is equal to the intrinsic torsion of an Aristotelian geometry [36]. Intrinsic torsion loosely speaking is a torsion tensor that is constructed from the geometric data $\tau_\mu$ and $h_{\mu\nu}$ that is first order in derivatives.

The Riemann tensor associated with this affine connection is

$$R_{\mu\nu\sigma}{}^{\rho} = -\partial_{\mu}\Gamma^{\rho}_{\nu\sigma} - \Gamma^{\rho}_{\mu\lambda}\Gamma^{\lambda}_{\nu\sigma} - (\mu \leftrightarrow \nu). \tag{41}$$

A straightforward calculation tells us that

$$R_{\mu\nu\sigma}{}^{\rho} = \epsilon^{bc}e^{\rho}{}_{b}e_{\sigma c}\left(\partial_{\mu}\omega_{\nu} - \partial_{\nu}\omega_{\mu}\right). \tag{42}$$

The Ricci tensor is defined as $R_{\mu\sigma} = R_{\mu\rho\sigma}{}^{\rho}$. It follows that

$$h^{\mu\sigma}R_{\mu\sigma} = \epsilon^{bc}e^{\rho}{}_{b}e^{\mu}{}_{c}\left(\partial_{\mu}\omega_{\rho} - \partial_{\rho}\omega_{\mu}\right). \tag{43}$$

## 3.3  Fracton gauge fields on an Aristotelian geometry

From the torsion constraint $de^{a} - \epsilon^{ab}\omega \wedge e_{b} = 0$ we can deduce (by applying the exterior differential) that

$$d^{2}e^{a} - \epsilon^{ab}d\omega \wedge e_{b} + \epsilon^{ab}\omega \wedge de_{b} = 0 \qquad \Leftrightarrow \qquad e^{a} \wedge d\omega = 0. \tag{44}$$

Using this it can be shown that the first term on the last line of (33) can be written as

$$a \wedge d\omega = \phi\tau \wedge d\omega = \frac{\phi}{2}e^{\rho}{}_{a}e^{\sigma}{}_{b}\left(\partial_{\rho}\omega_{\sigma} - \partial_{\sigma}\omega_{\rho}\right)\tau \wedge e^{a} \wedge e^{b} = -\frac{\phi}{2}h^{\mu\nu}R_{\mu\nu}\tau \wedge e^{1} \wedge e^{2}, \tag{45}$$

where $\phi$ is given by $\phi = n^{\mu}a_{\mu}$. In other words we decompose the gauge potential $a_{\mu}$ as

$$a_{\mu} = \phi\tau_{\mu} + \phi_{a}e_{\mu}{}^{a}, \tag{46}$$

where $\phi = n^{\mu}a_{\mu}$ and $\phi_{a} = e^{\mu}{}_{a}a_{\mu}$. In order to rewrite the second term on the last line of (33) we use that

$$K_{ab} = e^{\mu}{}_{a}e^{\nu}{}_{b}K_{\mu\nu} = \frac{1}{2}n^{\rho}e^{\sigma}{}_{a}\left(\partial_{\rho}e_{\sigma b} - \partial_{\sigma}e_{\rho b}\right) + (a \leftrightarrow b). \tag{47}$$

Using this we can write

$$de^{a} \wedge e^{c}\epsilon_{ab}S^{b}{}_{c} = K^{ad}S^{bc}\left(\delta_{ad}\delta_{bc} - \delta_{bd}\delta_{ac}\right)\tau \wedge e^{1} \wedge e^{2}. \tag{48}$$

The last term on the last line of (33) can be written as

$$\tau \wedge d\tau = \frac{1}{2}\epsilon^{ab}e^{\rho}{}_{a}e^{\sigma}{}_{b}\left(\partial_{\rho}\tau_{\sigma} - \partial_{\sigma}\tau_{\rho}\right)\tau \wedge e^{1} \wedge e^{2} = \frac{1}{2}\varepsilon^{\rho\sigma\kappa}\tau_{\kappa}\left(\partial_{\rho}\tau_{\sigma} - \partial_{\sigma}\tau_{\rho}\right)\tau \wedge e^{1} \wedge e^{2}, \tag{49}$$

where $\varepsilon^{\rho\sigma\kappa} = e^{-1}\epsilon^{\rho\sigma\kappa}$ with $\epsilon^{\rho\sigma\kappa}$ the Levi-Civita symbol and $e = \det(\tau_{\mu}, e_{\mu}{}^{a})$. Hence, we obtain the following expression for the Lagrangian (33)

$$L_{\text{CS}} = \left(-\mu\phi h^{\mu\nu}R_{\mu\nu} - 4\mu K^{ad}S^{bc}\delta_{d[a}\delta_{b]c} + \frac{\mu_{H}}{2}\varepsilon^{\rho\sigma\kappa}\tau_{\kappa}\left(\partial_{\rho}\tau_{\sigma} - \partial_{\sigma}\tau_{\rho}\right)\right)\tau \wedge e^{1} \wedge e^{2}. \tag{50}$$

Let us define the following symmetric tensor

$$A_{\mu\nu} = e_{\mu}{}^{a}e_{\nu}{}^{b}S_{ab} = A_{\rho}{}^{a}P^{\rho}_{(\mu}e_{\nu)a}, \tag{51}$$

where $P^{\rho}_{\mu} = h_{\mu\nu}h^{\nu\rho} = \delta^{\rho}_{\mu} - \tau_{\mu}n^{\rho}$. We can then finally write the action as

$$\begin{aligned} S[\phi, A_{\mu\nu}, \tau_{\mu}, h_{\mu\nu}] = \int d^{3}x\, e\Big(&-\mu\phi h^{\mu\nu}R_{\mu\nu} + 2\mu K_{\mu\rho}A_{\nu\sigma}\left(h^{\mu\nu}h^{\rho\sigma} - h^{\mu\rho}h^{\nu\sigma}\right) \\ &+ \frac{\mu_{H}}{2}\varepsilon^{\rho\sigma\kappa}\tau_{\kappa}\left(\partial_{\rho}\tau_{\sigma} - \partial_{\sigma}\tau_{\rho}\right)\Big). \end{aligned} \tag{52}$$

This is a gauge invariant coupling of an Aristotelian geometry to fracton gauge fields. We will refer to this action as the second order formulation of the theory given in (10).

We will next consider the gauge symmetries of this theory. In particular, with a generalization to higher dimensions in mind, we will try to understand it independently of its Chern–Simons formulation. We will use the $\bar{\delta}A_\mu$ transformations of (25) which we repeat here are defined as

$$\bar{\delta}A_\mu = \mathcal{L}_\xi A_\mu + \partial_\mu \Sigma + \left[A_\mu, \Sigma\right], \tag{53}$$

where $\Sigma$ is given by (26). In components we have

$$\bar{\delta}a_\mu = \mathcal{L}_\xi a_\mu + \partial_\mu \bar{\Lambda} + e_\mu{}^a \bar{\Lambda}^a. \tag{54}$$

Likewise, we have

$$\bar{\delta}e_\mu{}^a = \mathcal{L}_\xi e_\mu{}^a + \bar{\lambda}\epsilon^a{}_b e_\mu{}^b, \tag{55}$$

as well as

$$\bar{\delta}A_\mu{}^a = \mathcal{L}_\xi A_\mu{}^a + \partial_\mu \bar{\Lambda}^a + \bar{\lambda}\epsilon^a{}_b A_\mu{}^b - \omega_\mu \epsilon^a{}_b \bar{\Lambda}^b. \tag{56}$$

For the clock form $\tau$ we can write

$$\bar{\delta}\tau_\mu = \mathcal{L}_\xi \tau_\mu. \tag{57}$$

Using the definition of the inverse vielbeine (24) we find

$$\bar{\delta}e^\mu{}_a = \mathcal{L}_\xi e^\mu{}_a + \bar{\lambda}\epsilon_a{}^b e^\mu{}_b, \tag{58a}$$

$$\bar{\delta}n^\mu = \mathcal{L}_\xi n^\mu. \tag{58b}$$

The field $\phi_a$ does not enter the action. In fact we can gauge-fix it to be zero. This is because we have

$$\bar{\delta}\phi_a = \delta(e^\mu{}_a a_\mu) = e^\mu{}_a \partial_\mu \bar{\Lambda} + \bar{\Lambda}_a + \bar{\lambda}\epsilon_a{}^b \phi_b + \xi^\mu \partial_\mu \phi_a, \tag{59}$$

so that for $\bar{\Lambda}_a = -e^\mu{}_a \partial_\mu \bar{\Lambda}$, we can set $\phi_a = 0 = \bar{\delta}\phi_a$.

The diffeomorphisms and gauge transformation for the remaining second-order fields entering the action are

$$\bar{\delta}\phi = \bar{\delta}\left(n^\mu a_\mu\right) = \mathcal{L}_\xi \phi + n^\mu \partial_\mu \bar{\Lambda}, \tag{60}$$

and

$$\bar{\delta}A_{\mu\nu} = -P^\rho_{(\mu} P^\sigma_{\nu)} \nabla_\rho \partial_\sigma \bar{\Lambda}, \tag{61}$$

where we used $\bar{\Lambda}_a = -e^\mu{}_a \partial_\mu \bar{\Lambda}$.

Finally we will verify that the second order action is gauge invariant with respect to the $\bar{\Lambda}$ gauge transformation. If we take the second order action (52) and vary it with respect to $\bar{\Lambda}$, i.e., using (60) and (61), then after performing a few partial integrations and using the identity (B.9),[4] we end up with

$$\delta_{\bar{\Lambda}} S = -2\mu \int d^3 x\, e\, \bar{\Lambda} h^{\kappa\rho} h^{\lambda\sigma} K_{\kappa\lambda} \left(R_{\rho\sigma} - \frac{1}{2} h_{\rho\sigma} h^{\alpha\beta} R_{\alpha\beta}\right). \tag{63}$$

We can see that this identically zero since

$$h^{\kappa\rho} h^{\lambda\sigma} \left(R_{\rho\sigma} - \frac{1}{2} h_{\rho\sigma} h^{\alpha\beta} R_{\alpha\beta}\right) = 0, \tag{64}$$

following from (42).

---

[4]Since the connection has torsion it is useful to note the following when performing partial integrations

$$\nabla_\mu X^\mu = e^{-1} \partial_\mu (e X^\mu) + T^\mu_{\mu\nu} X^\nu, \tag{62}$$

for any vector $X^\mu$.

If we add the cosmological constant term of Section 2.4 and go to the second order formulation we end up with

$$S[\phi, A_{\mu\nu}, \tau_\mu, h_{\mu\nu}] = \int d^3x \; e \left( -\mu\phi(h^{\mu\nu}R_{\mu\nu} - 2\Lambda) + 2\mu K_{\mu\rho}A_{\nu\sigma}(h^{\mu\nu}h^{\rho\sigma} - h^{\mu\rho}h^{\nu\sigma}) \right.$$
$$\left. + \frac{\mu_H}{2}\varepsilon^{\rho\sigma\kappa}\tau_\kappa (\partial_\rho\tau_\sigma - \partial_\sigma\tau_\rho) \right). \tag{65}$$

In summary, we have brought the Chern–Simons theory based on the fracton algebra, with and without cosmological constant, from the first order to the second order formulation. In the second order formulation, the Lagrangian depends on the geometric quantities that describe an Aristotelian geometry, $(\tau_\mu, h_{\mu\nu})$. This is analogous to the reformulation of Chern–Simons theories based on the Poincaré or (A)dS algebras to three-dimensional gravity in the metric formulation [37, 38], but without Lorentzian boost symmetry. Additionally, our theory naturally incorporates the coupling of a fractonic electromagnetic field, described by $(\phi, A_{\mu\nu})$ to the Aristotelian gravitational theory, such that we have a generalization of dipole conservation to curved and unrestricted Aristotelian geometry. Further physical implications of this theory are discussed in [26].

### 3.4 Generalization to higher dimensions

It is only in the last step, equation (64), that we explicitly use that we are in $2 + 1$ dimensions. One of the benefits of the second order formulation (52) is that it can be straightforwardly generalized to higher dimensions.

Explicitly, if we take the action

$$S[\phi, A_{\mu\nu}, \tau_\mu, h_{\mu\nu}] = \int d^{d+1}x \; e \left( -\mu\phi h^{\mu\nu}R_{\mu\nu} + 2\mu K_{\mu\rho}A_{\nu\sigma}(h^{\mu\nu}h^{\rho\sigma} - h^{\mu\rho}h^{\nu\sigma}) \right) + S_{\tau,h}, \tag{66}$$

where all fields are now defined in $d + 1$ dimensions, then if we modify the gauge transformation of $A_{\mu\nu}$ under $\bar\Lambda$ to

$$\bar\delta\phi = n^\mu\partial_\mu\bar\Lambda, \tag{67a}$$

$$\bar\delta A_{\mu\nu} = -P^\rho_{(\mu}P^\sigma_{\nu)}\left[ \nabla_\rho\partial_\sigma\bar\Lambda - \bar\Lambda\left( G_{\rho\sigma} - \frac{1}{d-1}h_{\rho\sigma}h^{\kappa\lambda}G_{\kappa\lambda} \right) \right], \tag{67b}$$

in which we defined

$$G_{\mu\nu} = R_{\mu\nu} - \frac{1}{2}h_{\mu\nu}h^{\alpha\beta}R_{\alpha\beta}, \tag{68}$$

whose spatial projection is a $d$-dimensional Einstein tensor, it follows that (66) is gauge invariant under the $\bar\Lambda$ transformation. Note that in (66) we left out the term proportional to $\mu_H$ in (52). This is because that term does not generalise so straightforwardly to higher dimensions. We replaced it with the action $S_{\tau,h}$ which only depends on the fields $\tau_\mu$ and $h_{\mu\nu}$. For example we can take for the action $S_{\tau,h}$ the following [39],

$$S_{\tau,h} = \frac{1}{4}\int d^{d+1}x \; e \, h^{\mu\rho}h^{\nu\sigma}(\partial_\mu\tau_\nu - \partial_\nu\tau_\mu)(\partial_\rho\tau_\sigma - \partial_\sigma\tau_\rho). \tag{69}$$

In principle we could take a Hořava–Lifshitz type action for $S_{\tau,h}$ whose diffeomorphism invariant formulation can be given in terms of Aristotelian geometry [40].

If we are in $2+1$ dimensions, i.e., we consider (66) for $d=2$ with (69), then we know from the rewriting of the first order action that the $\tau_\mu$ equation of motion (upon using all the other equations of motion) cannot receive any contributions from the terms in (66) that are proportional to $\mu$. If vary (66) in which we use (69) with respect to $\tau_\mu$ we get that $\tau_\mu$ must obey [39],

$$h^{\mu\rho}h^{\nu\sigma}\left(\partial_\mu\tau_\nu-\partial_\nu\tau_\mu\right)\left(\partial_\rho\tau_\sigma-\partial_\sigma\tau_\rho\right)=0\,, \tag{70}$$

which is equivalent (in form notation) to $\tau\wedge d\tau=0$. To get this result it is sufficient to vary $\tau_\mu$ as $\delta\tau_\mu=\Omega\tau_\mu$ where $\Omega$ is an arbitrary function (while keeping $h_{\mu\nu}$ fixed). This says that $\tau$ must be hypersurface orthogonal which is less constraining than what we had for the 3D CS theory in which we found that $d\tau=0$. The variation of $S_{\tau,h}$ with respect to $h_{\mu\nu}$ vanishes upon using the condition (70). It would be interesting to work out the equations of motion of (66) with (69) in general dimensions.

We can generalize (66) by adding a cosmological constant. The action becomes

$$S_\Lambda=\int d^{d+1}x\ e\left(-\mu\phi(h^{\mu\nu}R_{\mu\nu}-2\Lambda)+2\mu K_{\mu\rho}A_{\nu\sigma}\left(h^{\mu\nu}h^{\rho\sigma}-h^{\mu\rho}h^{\nu\sigma}\right)\right)+S_{\tau,h}\,, \tag{71}$$

where the only modification is the appearance of a "cosmological constant" term $e\phi\Lambda$, with $\Lambda=\sigma\frac{d(d-1)}{2\ell^2}$ where $\sigma=-1,1$ and $\ell$ is a length (see Appendix C). The fracton gauge transformations are then modified to

$$\bar{\delta}\phi=n^\mu\partial_\mu\bar{\Lambda}\,, \tag{72a}$$

$$\delta A_{\mu\nu}=-P^\rho_{(\mu}P^\sigma_{\nu)}\left[\nabla_\rho\partial_\sigma\bar{\Lambda}-\left(G_{\rho\sigma}-\frac{1}{d-1}h_{\rho\sigma}h^{\alpha\beta}G_{\alpha\beta}-\frac{1}{d-1}h_{\rho\sigma}\Lambda\right)\bar{\Lambda}\right]. \tag{72b}$$

All theories (in three or higher spacetime dimensions, with and without cosmological constant) share many similarities with magnetic Carroll gravity defined in [41] and studied, e.g., in [42, 43]. Besides the issue of interpreting the different fields entering the action, the main difference between these two physical situations lies mainly in the treatment of the clock form and the issue of boost-invariance. While magnetic Carroll gravity is a boost-invariant theory for the Carrollian metric, the equivalent gauge-invariance in fractonic theories has been exploited to arrive at the transformation laws (61) (or (67b) or (72b)). The clock form in Carroll gravity is a dynamical object while here it is a fixed part of the geometry, subject to the constraints obtained by variation of $S_{\tau,h}$.

## 3.5 Coupling to matter

In [11, 12] it was shown that the complex scalar field $\Phi$ with global dipole symmetry can be coupled to an arbitrary curved Aristotelian geometry leading to the following action (where we adapted the result of [11] to the notation used here)

$$S_{\text{scalar}}=\int d^{d+1}x\ e\left[\left(n^\mu\partial_\mu\Phi-i\phi\Phi\right)\left(n^\nu\partial_\nu\Phi^\star+i\phi\Phi^\star\right)-m^2|\Phi|^2-\lambda h^{\mu\nu}h^{\rho\sigma}\hat{X}_{\mu\rho}\hat{X}^\star_{\nu\sigma}\right], \tag{73}$$

where

$$\hat{X}_{\mu\nu}=P^\rho_{(\mu}P^\sigma_{\nu)}\left(\partial_\rho\Phi\partial_\sigma\Phi-\Phi\nabla_\rho\partial_\sigma\Phi\right)-iA_{\mu\nu}\Phi^2\,, \tag{74}$$

in which $\nabla_\rho$ is covariant with respect to the Aristotelian connection (36). The parameters $m^2$ and $\lambda$ are real numbers.

The Lagrangian (73) is gauge invariant under the gauge transformations

$$\delta\phi=n^\mu\partial_\mu\bar{\Lambda}\,,\qquad\delta A_{\mu\nu}=-P^\rho_{(\mu}P^\sigma_{\nu)}\nabla_\rho\partial_\sigma\bar{\Lambda}\,,\qquad\delta\Phi=i\bar{\Lambda}\Phi\,. \tag{75}$$

Comparing (72a) and (72b) with (75) we see that the coupling to (73) only works in $2+1$ dimensions (with $\Lambda = 0$). We can now simply add the actions (66) and (73) and set $d = 2$ leading to

$$
\begin{aligned}
S = \int d^3x \; e \Big( -\mu \phi h^{\mu\nu} R_{\mu\nu} + 2\mu K_{\mu\rho} A_{\nu\sigma} (h^{\mu\nu} h^{\rho\sigma} - h^{\mu\rho} h^{\nu\sigma}) \\
+ \left( n^\mu \partial_\mu \Phi - i\phi\Phi \right) (n^\nu \partial_\nu \Phi^\star + i\phi\Phi^\star) - m^2 |\Phi|^2 - \lambda h^{\mu\nu} h^{\rho\sigma} \hat{X}_{\mu\rho} \hat{X}^\star_{\nu\sigma} \Big) + S_{\tau,h} ,
\end{aligned}
\tag{76}
$$

where $S_{\tau,h}$ is for example given by (69) or by the term proportional to $\mu_H$ in (52).

If we now vary $\phi$ and $A_{\mu\nu}$ we get equations for the curvature that are determined by the scalar field. Hence, this theory is not necessarily restricted to maximally symmetric spacetimes (as was the case in [9,11] where the dipole gauge theories were quadratic in the gauge fields).

## 4 Solutions and charges

In this Section we will derive circularly symmetric solutions of the Chern–Simons theory (for any $\Lambda$). Geometrically they share similarities with the spatial geometries of $2+1$ dimensional gravity, but our analysis of the charges shows that they carry electric charge and we therefore interpret them as monopoles. We also comment on asymptotic symmetries which infinitely enhance the fracton algebra.

### 4.1 Circularly symmetric solutions in $2+1$ dimensions

In this section, we will discuss circularly symmetric solutions to the field equations (23), which describe the field generated by an electric monopole in a curved background with and without a cosmological constant.

We will make use of the first order formulation. Let us consider a circularly symmetric ansatz of the form

$$
\tau = N(r)dt , \quad e^1 = \frac{1}{f(r)} dr , \quad e^2 = rd\theta , \quad a = \phi(r)dt , \quad S_{ab} = S(r)\delta_{ab} ,
\tag{77}
$$

where $\theta$ is $2\pi$ periodic. We only specify $S_{ab}$, which we defined as $A^a =: \tilde{A}^a + S^{ab} e^b$, since $\tilde{A}$ will be determined by $a$ (as we have already discussed around (32)).

The Aristotelian geometry given by $\tau$ and $e^a$ is completely determined by (23a), (23c) and (23e). Indeed, the equation of motion (23a) shows that $N(r)$ is constant. In particular, by selecting an appropriate time normalization, $N$ can be set to one, resulting in

$$
\tau = dt .
\tag{78}
$$

Additionally, equations (23c) and (23e) imply

$$
\omega = f(r) d\theta ,
\tag{79}
$$

with

$$
f(r) = \sqrt{-\Lambda r^2 - M} ,
\tag{80}
$$

where $M$ is a real constant.[5]

---

[5]Strictly speaking there is the freedom to have both signs, i.e., $f(r) = \pm\sqrt{-\Lambda r^2 - M}$, but since we can absorb this freedom into the orientation of $\theta$ in (79) we will restrict henceforth to the positive root.

The fractonic fields are determined by (23b) and (23d). One finds

$$\phi(r) = \phi_0 \sqrt{-\Lambda r^2 - M}, \qquad S(r) = S_0, \tag{81}$$

where $\phi_0$ and $S_0$ are integration constants and $A^a = \delta_1^a \Lambda r \phi_0 dt + S_0 e^a$.

In sum, the Aristotelian geometry of the circularly symmetric solution is described by the following clock form and spatial metric:

$$\tau = dt, \qquad h_{\mu\nu} dx^\mu dx^\nu = \frac{dr^2}{-\Lambda r^2 - M} + r^2 d\theta^2, \tag{82}$$

while the fractonic fields are given by (cf. (51))

$$\phi = \phi_0 \sqrt{-\Lambda r^2 - M}, \qquad A_{\mu\nu} = S_0 h_{\mu\nu}. \tag{83}$$

In analogy to their lorentzian geometries we called the integration constant $M$, but it should not be interpreted as a mass, but rather as a charge. This can be inferred from the fact that the curvature of the geometry (23c) comes from the coupling $a_\mu J^\mu$ rather than from coupling to $e_\mu^a$.

Let us first focus on the flat case, which is the well-defined limit $\Lambda \to 0$ with metric $-\frac{dr^2}{M} + r^2 d\theta^2$ (for the following remarks further details are, e.g., in [44,45] and references therein). For $M = -1$ this is the plane with flat metric, while for $-1 < M < 0$ the plane is deformed into a cone, which is metrically flat except at the tip which can be interpreted as a point particle. When $M \to 0$ the geometry approaches a cylinder and when $M < -1$ it is a conical excesses. When $M > 0$ we can think about it as a Milne universe. The ansatz in (77) assumes a static and circularly symmetric configuration, thereby precluding the possibility of deriving a rotating solution from it. In relativistic gravitational theories, a common technique to obtain rotating solutions involves applying an improper boost to a static and circularly symmetric solution. However, this method is not applicable here due to the absence of boosts. Nevertheless, one could consider a more general ansatz that is stationary and invariant under the Killing vector $\partial_\theta$, allowing the metric to include cross terms with $dt d\theta$. We will explore this possibility in the future.

Let us from now on focus on $\Lambda < 0$ where the spatial metric takes precisely the same form as the spatial metric of a nonrotating BTZ black hole [46, 47] in general relativity in $2 + 1$ dimensions. However, the clock form is different since the clock form does not depend on any integration constant of the spatial geometry as would have been the case for the lorentzian geometry. Furthermore, the geometry does not depend on the integration constants $\phi_0$ and $S_0$ of the fractonic gauge fields, meaning there is no backreaction of the gauge fields on the geometry. This is similar to the case of Einstein gravity in $2 + 1$ dimensions coupled to $U(1)$ abelian Chern-Simons fields (see, e.g., [48]).

The gauge connection associated with this solution is given by

$$A_t = H + \phi_0 \sqrt{-\Lambda r^2 - M} Q + \Lambda r \phi_0 D_1, \tag{84a}$$

$$A_r = \frac{1}{\sqrt{-\Lambda r^2 - M}} P_1 + \frac{S_0}{\sqrt{-\Lambda r^2 - M}} D_1, \tag{84b}$$

$$A_\theta = r P_2 + \sqrt{-\Lambda r^2 - M} J + r S_0 D_2. \tag{84c}$$

In complete analogy with the Chern-Simons formulation of Einstein gravity, it is possible to gauge away all the dependence on the radial coordinate $r$, such that the physical information is encoded in an auxiliary connection $\mathfrak{a} = \mathfrak{a}_t dt + \mathfrak{a}_\theta d\theta$, where $A = h^{-1}(d + \mathfrak{a})h$ for some gauge group element $h$. For the circularly symmetric solution, one explicitly finds that

$$h = \exp\left[\frac{1}{\sqrt{-\Lambda}} \text{Arcoth}\left(\sqrt{1 + \frac{M}{\Lambda r^2}}\right)(P_1 + S_0 D_1)\right], \tag{85}$$

where the auxiliary connection is given by

$$\mathfrak{a} = \left(H + \sqrt{-M}\phi_0 Q\right)dt + \sqrt{-M}J d\theta. \tag{86}$$

The removal of the radial dependence via (85) can only be achieved for negative values of $\Lambda$ and $M$. For vanishing cosmological constant the auxiliary connection takes exactly the same form as (86) and the group element $h$ simplifies to $h = \exp\left[\frac{r}{\sqrt{-M}}\left(P_1 + S_0 D_1\right)\right]$.

## 4.2 Charges and asymptotic symmetries

The charges of this theory are related to large gauge transformations. To determine them we need to find gauge transformations that preserve the form of the auxiliary connection (86), i.e., we must find an $\varepsilon$ such that $d\varepsilon + [\mathfrak{a}, \varepsilon] = 0$ and which leads to non-vanishing charges. The charge associated with these large gauge transformations can then be obtained using the canonical formalism [49] and it is given by the following expression [50]

$$\delta\mathcal{Q}[\varepsilon] = -2\oint d\theta \langle \varepsilon \delta \mathfrak{a}_\theta \rangle. \tag{87}$$

When a transformation changes the charge it should not be thought of as a nonphysical gauge redundancy, but as an observable physical change.

For the case at hand large gauge transformations are generated by

$$\varepsilon = \bar{\Lambda}Q, \tag{88}$$

for a constant $\bar{\Lambda}$.[6] When $\bar{\Lambda}$ has no functional variation we find the electric charge

$$\mathcal{Q}[\varepsilon] = -4\pi\bar{\Lambda}\mu\sqrt{-M}. \tag{89}$$

This shows that the integration constant $M$, which geometrically shares some similarities with mass (but is not the mass of the system), is associated to the electric charge of the system.

The total energy can be obtained by considering the charge associated with time evolution and can be derived from

$$\delta E = 2\oint d\theta \langle \mathfrak{a}_t \delta \mathfrak{a}_\theta \rangle. \tag{90}$$

Then, if one assumes that $\delta\phi_0 = 0$, then the energy of the solution takes the form

$$E = -2\pi\mu\phi_0 M. \tag{91}$$

Therefore the total energy and the electric charge are related $E \sim \phi_0 \mathcal{Q}^2$. The constant $S_0$ does not appear in the charges. Indeed, since $A_{\mu\nu} = S_0 h_{\mu\nu}$, the constant $S_0$ can be interpreted as as labeling a particular ground state of the symmetric tensor $A_{\mu\nu}$.

Note that the components of the auxiliary connection (86) are defined along the generators $H$, $J$ and $Q$, which form a set of commuting generators. This suggests that a natural set of asymptotic conditions that accommodate this solution could be given by "soft hairy asymptotic conditions", similar to those introduced in [51–53] whose asymptotic symmetry algebra is given by a set of $U(1)$ Kac-Moody current algebras. This aligns with the fact that the dipole algebra with a negative cosmological constant is isomorphic, apart from the central element $H$, to the three-dimensional Poincaré algebra. Indeed, this isomorphism allows us to map all

---

[6]When $\chi_J$ is nonzero there are more large general transformations that lead to non-vanishing charges, but they will not provide additional information since they will be proportional to this charge.

the known results in three-dimensional general relativity in flat space to the case of the dipole algebra with a negative cosmological constant. Indeed, based on the results in [54], after a suitable gauge transformation, it would be possible to write a set of asymptotic conditions in the flat space analogue of the highest weight gauge [55–57], where

$$a_\theta = (J - \ell P_1) - \frac{\mathcal{M}(t,\theta)}{2}(J + \ell P_1) - \frac{\mathcal{L}(t,\theta)}{2}\left(Q + \frac{1}{\ell}D_2\right) - \frac{\mathcal{H}(t,\theta)}{2}H\,. \qquad (92)$$

Here $\ell$ is the *AdS* radius related to the cosmological constant by $\Lambda = -\ell^{-2}$. The asymptotic symmetry algebra is then given by the three-dimensional BMS algebra with an additional $U(1)$ current, which contains the cosmological dipole algebra as its wedge algebra. The case when the cosmological constant vanishes is less clear, as the previous asymptotic conditions do not appear to have a natural flat limit. However, (86) suggests that soft hairy asymptotic conditions might naturally be applicable. We plan to investigate this further in the future.

## 5  Discussion

We conclude by recalling that the main result of the paper is a metric formulation of a fracton gauge theory (52) obtained from a Chern-Simon action in 2+1 dimensions, with gauge fields $A_{\mu\nu}$ and $\phi$ coupled to dynamical Aristotelian gravitation fields $h_{\mu\nu}$ and $\tau_\mu$. We want to emphasize that the invariance of the action under the dipole gauge transformations given by Eqs. (60) and (61) imposes no restrictions on the geometry, in stark contrast to previous no-go results. [9, 11, 12]. This theory was generalized to higher dimensions without (66) and with (71) a cosmological constant. Additionally, we showed that the three-dimensional theory can be consistently coupled to fractonic matter fields (73).

This work opens various interesting avenues for further exploration:

**Other multipole symmetries**    The tools we have used in this work are of course not restricted to dipole symmetries and it could be interesting to generalize to higher multipole moments.

**Supersymmetrization**    An immediate generalization is the supersymmetrization of the CS theory. Again using the correspondence to Carroll symmetries [11, 27, 28] it is clear that such a theory exists [58] and it could be interesting to generalize the work of Huang [26] and ours to this framework.

**Aristotelian black holes**    The circularly symmetric solution described in Section 4, with negative cosmological constant, shares many properties with the BTZ black hole in General Relativity. It is therefore natural to ask whether it is possible to define a notion of an "Aristotelian black hole". Given the significant differences between the properties of Aristotelian and Riemannian geometries, one might attempt to extend the concept of an event horizon to Aristotelian geometries, as well as their thermal properties, as was done, for example, in the case of Carrollian gravitational theories [59]. One possible approach is to leverage the isomorphism between the dipole algebra with a negative cosmological constant and the three-dimensional Poincaré algebra to describe thermal solutions within the Chern-Simons formulation of the Aristotelian theory. In particular, we would like to study whether the flat-space generalization of the Cardy formula [60, 61] could play a significant role in describing the thermal properties of Aristotelian black holes, as well as its connection to the BMS-type asymptotic conditions outlined in (92).

**Fracton BF gravity**    There also exist generalizations to (1+1)-dimensional gravitational models [62,63] in particular there is an analog proposal for fracton BF gravity [59] to which much of what we have done could be applied.

**Relation to scalar charge gauge theories**    In order to make contact with more standard gauge theories of fractons on flat space [13, 14] let us take the action (52) and choose a background configuration which satisfies the equations of motion at $0^{\text{th}}$-order, given by the flat Aristotelian background $\bar{\tau}_t = 1$ and $\bar{h}_{ij} = \delta_{ij}$ and all other fields are zero. Linearizing the theory (52) up to quadratic order around this background yields, among others, the term $E^{ij}\dot{A}_{ij} + \phi\,\partial_i\partial_j E^{ij}$ (where $E_{ij}$ is related to the metric perturbation $e_{ij} = h_{ij} - \bar{h}_{ij}$ by $E_{ij} = e_{ij} - \delta_{ij}e_k{}^k$) which is ubiquitous in the Hamiltonian treatment of fracton gauge theories performed, e.g., in [11]. We reserve a more thorough study of the Hamiltonian formulation of the theory displayed in (66) or (71) and its relation to the theories (or others) described in [11, 13, 14] for future works.

**Infrared triangle, memory effects**    It was recently shown [33,34] that fracton theories allow for interesting interrelations, called infrared triangle [64], between asymptotic symmetries, soft theorems and (double kick) memory effects. The fracton CS theory also allows for infinite dimensional asymptotic symmetries (cf., Section 4.2) which makes it natural to expect related soft theorems and memory effects. What makes the case at hand an interesting challenge is that the gauge theory is topological and therefore has no propagating degrees of freedom.

**Applications to condensed matter systems**    We briefly suggest a possible application of our fractonic Chern-Simons theory in the framework of topological phases of matter. We first observe that our action (22) can be seen as an one-loop effective topological field theory induced by integrating out some massive degrees of freedom. In particular, in a $(2+1)$-dimensional microscopic system made by non-relativistic massive fermions with conserved electric charge, dipole and rotational symmetry but broken time-reversal symmetry, (22) can describe the topological response of the system to external probings. In fact, the first term is known in the condensed-matter literature as first Wen-Zee term [65] and in absence of dipole conservation, it has been employed to study several kinds of topological systems, such as quantum Hall insulators and higher-order topological phases in two space dimensions [66–70]. The *first Wen-Zee term*, entirely related to the charge conservation and rotational symmetry of the system, gives rise to the shift invariant and corresponding Hall viscosity. On the other hand, the first term related to the cosmological constant $\Lambda$ in the same action coincides with the topological response of an atomic insulator in two space dimensions [71], which only depends on charge conversation and translation symmetry. Finally, the first term related to $\chi_J$ in (22) is known as *second Wen-Zee term* and plays a role mainly in the fractional quantum Hall effect [72,73]. Thus, we expect that our $(2+1)$-dimensional fractonic theory with a non-zero cosmological constant represents the low-energy description of suitable topological phases augmented by the dipole symmetry, namely topological dipole phases (see [74] for an example of topological dipole insulator) that will be investigated in detail in a future work.

# Acknowledgments

We acknowledge support from the Erwin Schrödinger International Institute for Mathematics and Physics (ESI) where some of the research was undertaken at the "Carrollian Physics and Holography" thematic programme.

**Funding information** This work was initiated while StP was supported by the Leverhulme Trust Research Project Grant (RPG-2019-218) "What is Non-Relativistic Quantum Gravity and is it Holographic?". JH was supported by the Royal Society University Research Fellowship Renewal "Non-Lorentzian String Theory" (grant number URF\R\221038). The work of SiP was partially supported by the Fonds de la Recherche Scientifique - FNRS under Grant No. FC.36447. SiP acknowledges the support of the SofinaBoël Fund for Education and Talent and the *Fonds Friedmann* run by the *Fondation de l'École polytechnique*. The research of AP is partially supported by Fondecyt grants No 1211226, 1220910 and 1230853.

# A  Affine connection

The purpose of this appendix is to show that equation (36) follows from (35) in which we substitute (29a) and (29b).

First we use completeness to write (35) as

$$\Gamma^\rho_{\mu\nu} = n^\rho \partial_\mu \tau_\nu + e^\rho{}_b \partial_\mu e_\nu{}^b - \epsilon^{bc} e^\rho{}_b e_{\nu c} \left( \tau_\mu n^\sigma \omega_\sigma + e_\mu{}^d e^\sigma{}_d \omega_\sigma \right) . \tag{A.1}$$

In this equation we substitute (29a) and (29b) leading to (after some straightforward algebra)

$$\Gamma^\rho_{\mu\nu} = n^\rho \partial_\mu \tau_\nu + h^{\rho\sigma} \tau_\mu K_{\nu\sigma} - h^{\rho\sigma} \tau_\mu h_{\sigma\kappa} \partial_\nu n^\kappa + X^\rho_{\mu\nu}, \tag{A.2}$$

where

$$X^\rho_{\mu\nu} = h^{\rho\lambda} h^{\kappa\sigma} \left( e_{\lambda b} h_{\sigma\mu} \partial_\kappa e_\nu{}^b + e_{\mu b} h_{\sigma\nu} \left( \partial_\kappa e_\lambda{}^b - \partial_\lambda e_\kappa{}^b \right) \right) . \tag{A.3}$$

Using completeness once more we can write

$$X^\rho_{\mu\nu} = \tau_\nu n^\alpha X^\rho_{\mu\alpha} + P^\alpha_\nu X^\rho_{\mu\alpha}, \tag{A.4}$$

where we have

$$X^\rho_{\mu\alpha} n^\alpha = -P^\kappa_\mu P^\rho_\alpha \partial_\kappa n^\alpha . \tag{A.5}$$

In order to rewrite $P^\alpha_\nu X^\rho_{\mu\alpha}$ we use that

$$h^{\rho\lambda} P^\kappa_\nu P^\sigma_\mu \left[ e_\sigma{}^b \left( \partial_\kappa e_\lambda{}^b - \partial_\lambda e_\kappa{}^b \right) + \text{cyclic permutations of } \sigma, \kappa, \lambda \right] = 0 . \tag{A.6}$$

Applying this identity to one-half of $X^\rho_{\mu\nu}$ while using (A.4) for the other half we find

$$X^\rho_{\mu\nu} = \tau_\nu X^\rho_{\mu\sigma} n^\sigma + \frac{1}{2} h^{\rho\lambda} P^\kappa_\nu P^\sigma_\mu \left( \partial_\kappa h_{\sigma\lambda} + \partial_\sigma h_{\lambda\kappa} - \partial_\lambda h_{\sigma\kappa} \right) . \tag{A.7}$$

After a bit of furthermore straightforward algebra we then find (36).

# B  Curvature

In this appendix we will collect some useful formulas for affine connections $\Gamma^\rho_{\mu\nu}$ with nonzero torsion.

The covariant derivative will be denoted by $\nabla_\mu$, the Riemann tensor by $R_{\mu\nu\sigma}{}^\rho$ and the torsion tensor by $T^\rho{}_{\mu\nu}$. The latter are defined via

$$\left[ \nabla_\mu, \nabla_\nu \right] X_\sigma = R_{\mu\nu\sigma}{}^\rho X_\rho - T^\rho{}_{\mu\nu} \nabla_\rho X_\sigma , \tag{B.1a}$$

$$\left[ \nabla_\mu, \nabla_\nu \right] X^\rho = -R_{\mu\nu\sigma}{}^\rho X^\sigma - T^\sigma{}_{\mu\nu} \nabla_\sigma X^\rho , \tag{B.1b}$$

from which it follows that

$$R_{\mu\nu\sigma}{}^{\rho} \equiv -\partial_{\mu}\Gamma^{\rho}_{\nu\sigma} + \partial_{\nu}\Gamma^{\rho}_{\mu\sigma} - \Gamma^{\rho}_{\mu\lambda}\Gamma^{\lambda}_{\nu\sigma} + \Gamma^{\rho}_{\nu\lambda}\Gamma^{\lambda}_{\mu\sigma}, \tag{B.2a}$$

$$T^{\rho}{}_{\mu\nu} \equiv 2\Gamma^{\rho}_{[\mu\nu]}. \tag{B.2b}$$

The algebraic and differential Bianchi identities are

$$R_{[\mu\nu\sigma]}{}^{\rho} = T^{\lambda}{}_{[\mu\nu}T^{\rho}{}_{\sigma]\lambda} - \nabla_{[\mu}T^{\rho}{}_{\nu\sigma]}, \tag{B.3}$$

$$\nabla_{[\lambda}R_{\mu\nu]\sigma}{}^{\kappa} = T^{\rho}{}_{[\lambda\mu}R_{\nu]\rho\sigma}{}^{\kappa}. \tag{B.4}$$

The Ricci tensor is defined as

$$R_{\mu\nu} \equiv R_{\mu\rho\nu}{}^{\rho}. \tag{B.5}$$

Our connection satisfies the property that

$$\Gamma^{\rho}_{\mu\rho} = \partial_{\mu}\log e, \tag{B.6}$$

where $e = \det(\tau_{\mu}, e^{a}_{\mu})$ is the integration measure. From this it follows that

$$R_{\mu\nu\rho}{}^{\rho} = 0. \tag{B.7}$$

The antisymmetric part of the Ricci tensor is then

$$2R_{[\mu\nu]} = T^{\lambda}{}_{\mu\nu}T^{\rho}{}_{\lambda\rho} + \nabla_{\mu}T^{\rho}{}_{\nu\rho} - \nabla_{\nu}T^{\rho}{}_{\mu\rho} + \nabla_{\rho}T^{\rho}{}_{\mu\nu}. \tag{B.8}$$

Consider the differential Bianchi identity (B.4) and contract $\kappa$ with $\nu$. Contracting the resulting identity with $n^{\lambda}h^{\mu\sigma}$ and using the torsion tensor (40) leads to

$$0 = \nabla_{\lambda}\left(n^{\lambda}h^{\mu\sigma}R_{\mu\sigma}\right) + 2h^{\kappa\sigma}h^{\rho\lambda}R_{\rho\sigma}K_{\kappa\lambda} \tag{B.9}$$
$$- 2\left(h^{\kappa\sigma}h^{\rho\lambda} - h^{\kappa\rho}h^{\lambda\sigma}\right)\left[\nabla_{\kappa}\nabla_{\rho}K_{\lambda\sigma} - \nabla_{\kappa}\left(K_{\lambda\sigma}\mathcal{L}_{n}\tau_{\rho}\right) + K_{\lambda\sigma}\mathcal{L}_{n}\tau_{\kappa}\mathcal{L}_{n}\tau_{\rho} - \nabla_{\rho}K_{\lambda\sigma}\mathcal{L}_{n}\tau_{\kappa}\right].$$

This identity is used in Section 3.3 to prove gauge invariance of the second order theory.

## C  Fracton algebraic considerations

We start by defining the (anti) de Sitter Carroll algebras in generic spacetime dimension $d+1$

$$\begin{aligned}
[J_{ab}, J_{cd}] &= \delta_{bc}J_{ad} - \delta_{ac}J_{bd} - \delta_{bd}J_{ac} + \delta_{ad}J_{bc}, & [B_a, P_b] &= \delta_{ab}H, \\
[J_{ab}, B_c] &= \delta_{bc}B_a - \delta_{ac}B_b, & [H, P_a] &= -\Lambda B_a, \\
[J_{ab}, P_c] &= \delta_{bc}P_a - \delta_{ac}P_b, & [P_a, P_b] &= -\Lambda J_{ab}.
\end{aligned} \tag{C.1}$$

where $\Lambda = \frac{\sigma}{\ell^2}$ with $\sigma = 1\ (-1)$ for (anti) de Sitter Carroll and is related to the cosmological constant $\frac{d(d-1)}{2}\Lambda$ (we follow [75,76]).

We now replace the Carroll boost by dipole moment $B_a \mapsto D_a$ and Carroll energy by charge $H \mapsto -Q$ and add a central element $H$ to the new algebra. This curved fracton algebra is then spanned by $\mathfrak{g} = \langle J_{ab}, H, P_a, Q, D_a \rangle$ and given by

$$\begin{aligned}
[J_{ab}, J_{cd}] &= \delta_{bc}J_{ad} - \delta_{ac}J_{bd} - \delta_{bd}J_{ac} + \delta_{ad}J_{bc}, & [P_a, D_b] &= \delta_{ab}Q, \\
[J_{ab}, D_c] &= \delta_{bc}D_a - \delta_{ac}D_b, & [Q, P_a] &= \Lambda D_a, \\
[J_{ab}, P_c] &= \delta_{bc}P_a - \delta_{ac}P_b, & [P_a, P_b] &= -\Lambda J_{ab}.
\end{aligned} \tag{C.2}$$

To obtain further geometric understanding let us think about the homogeneous space where we quotient $\mathfrak{g}$ by $\mathfrak{h} = \langle J_{ab}, Q, D_a \rangle$. The homogeneous space is a curved Aristotelian homogeneous space[7] either $\mathbb{R} \times \mathbb{S}^d$ or $\mathbb{R} \times \mathbb{H}^d$, for positive or negative $\Lambda$, respectively.

When we restrict to $2+1$ dimensions the rotations commute and with $J = -J_{12}$ we obtain the algebra

$$
\begin{aligned}
[J, P_a] &= \epsilon_{ab} P_b, & [J, D_a] &= \epsilon_{ab} D_b, & [P_a, D_b] &= \delta_{ab} Q, \\
[Q, P_a] &= \Lambda D_a, & [P_a, P_b] &= \Lambda \epsilon_{ab} J,
\end{aligned}
\tag{C.3}
$$

where $\epsilon_{12} = 1$. The most general invariant metric is

$$
\begin{aligned}
\langle J, Q \rangle &= \mu, & \langle P_a, D_b \rangle &= -\epsilon_{ab} \mu, & \langle H, H \rangle &= \mu_H, \\
\langle J, J \rangle &= \chi_J, & \langle P_a, P_b \rangle &= \chi_J \Lambda \delta_{ab},
\end{aligned}
\tag{C.4}
$$

which is nondegenerate for $\mu \neq 0 \neq \mu_H$, i.e., we are free to set $\chi_J$ to zero. The flat limit $\Lambda \to 0$ is well defined on the Lie algebra and invariant metric and consequentially also for the action. The main change is that we have the freedom to add an additional element $\langle J, H \rangle$ to the invariant metric which leads to

$$
\begin{aligned}
\langle J, Q \rangle &= \mu, & \langle P_a, D_b \rangle &= -\epsilon_{ab} \mu, & \langle H, H \rangle &= \mu_H, \\
\langle J, J \rangle &= \chi_J, & \langle J, H \rangle &= \chi_{JH}.
\end{aligned}
\tag{C.5}
$$

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
