# Peer review of "Fractons on curved spacetime in 2 + 1 dimensions"

_SciPost Physics, doi:SciPost Phys. 18, 022 (2025)_

## Round 1 · Referee Report · Mariano Cadoni (Referee 1) · 2024-11-4

Strengths

  1. The algebraic structure, the formal derivation and in general the whole mathematical setup is well described and organized.

Weaknesses

  1. Physical aspects and implications of the formalism are sometimes only sketched and not adequately discussed

Report

In this paper the authors discuss a non-relativistic version of Chern-Simon theory. The paper is well organized and clearly written. The results are of interest for several ares of gravitational physics an potentially may open the way to cross fertilization between the areas of gravitational and condensed matter physics. I recommend publication after the authors have considered my comments below.

Requested changes

  1. The equivalence between the Chern-Simon (CS) and geometric formulation of the theory should be discussed not only from the formal but also from the physical point of view. This is particularly needed because differently from usual CS theory they do not have the full relativistic symmetries so that the physical meaning of the 3D gravity theory (3.29) is not completely clear. Some points are addressed in Sect. (3.4), but it is not enough.

  2. As stated by the author there is an interesting relation between the circularly symmetric solutions of the theory and the BTZ black hole. This includes the notion of asymptotic symmetry, which in the relativistic case is at the heart of the AdS_3/CFT_2 correspondence (see the famous Brown & Henneaux paper) and of Strominger's microscopic derivation of the entropy of the BTZ black hole. It would be nice if the authors could at least comment on the possibility to extend this correspondence and related entropy computation to the nonrelativistic case under consideration.

Recommendation

Ask for minor revision

  • validity: good
  • significance: good
  • originality: high
  • clarity: good
  • formatting: good
  • grammar: excellent

Author:  Alfredo Perez  on 2024-11-25  [id 4986]

(in reply to Report 1 by Mariano Cadoni on 2024-11-04)

We sincerely thank the referee for their comments, which have undoubtedly contributed to improve the article and highlighting its physical content. To address these points, we have added two new paragraphs: one below Eq. (3.42) and another on page 17 in the discussion section of the revised version.

---

## Round 1 · Referee Report · Anonymous (Referee 2) · 2024-11-13

Strengths

1- Clear and well-defined goals, which are achieved 2- Excellent presentation 3- Topical research subject

Weaknesses

None

Report

The paper studies a specific Chern-Simons theory that has an interpretation in terms of fractons. While Chern-Simons theories are of course well-known and -studied, they offer a rich variety of physical systems depending on the choice of gauge group and boundary conditions, ranging from Quantum Hall physics to gravity. The present work picks an interesting intermediate spot that has both a gravity- and a cond-mat flavour. In particular, they chose the fracton algebra (2.1) as gauge algebra and proceed from there.

The techniques to study this theory are well-established, so in purely technical terms the papers offers no surprises. However, the novel and surprising aspect uncovered in their work are the specific features of the fracton gauge theory, its various extensions (e.g. to include a cosmological constant), and its asymptotic symmetry analysis.

The discussion section offers a perspective to further extensions and applications, and while one might lament that none such applications are addressed in this work, in my opinion the paper has enough meat to stand on its own, and its broad scope does not require specific examples of detailed applications.

In terms of research topic, style of presentation, and originality the paper is suitable for publication in SciPost Physics in its present form. Therefore, I suggest its publication in its present form.

Requested changes

None

Recommendation

Publish (easily meets expectations and criteria for this Journal; among top 50%)

  • validity: top
  • significance: high
  • originality: high
  • clarity: top
  • formatting: perfect
  • grammar: excellent

Author:  Alfredo Perez  on 2024-11-25  [id 4987]

(in reply to Report 2 on 2024-11-13)

We are grateful to the referee for their valuable comments and for recommending the article for publication in SciPost in its current form. To address the comments provided by Referee 1, we have added two new paragraphs: one below Eq. (3.42) and another on page 17 in the discussion section of the revised version.

---

## Round 2 · Referee Report · Mariano Cadoni (Referee 1) · 2024-11-26

Report

The authors have addressed the issues that I have raised, the paper is now suitable for publication

Recommendation

Publish (easily meets expectations and criteria for this Journal; among top 50%)

---

## Round 2 · Referee Report · Mariano Cadoni (Referee 1) · 2024-11-26

Report

The authors have addressed the issues I have raised , the paper is now suitable for publication.

Recommendation

Publish (easily meets expectations and criteria for this Journal; among top 50%)

---

## Round 2 · Author Response

We sincerely thank both referees for their comments. To address the comments provided by Referee 1, we have added two new paragraphs.

---

## Round 2 · List of Changes

We have incorporated the following two new paragraphs:

  • Below Eq. (3.42)

“In summary, we have brought the Chern--Simons theory based on the fracton algebra, with and without cosmological constant, from the first order to the second order formulation. In the second order formulation, the Lagrangian depends on the geometric quantities that describe an Aristotelian geometry, $(\tau_{\mu},h_{\mu\nu})$. This is analogous to the reformulation of Chern--Simons theories based on the Poincar\'e or (A)dS algebras to-three dimensional gravity in the metric formulation~\cite{Achucarro:1987vz,Witten:1988hc}, but without Lorentzian boost symmetry. Additionally, our theory naturally incorporates the coupling of a fractonic electromagnetic field, described by $(\phi,A_{\mu\nu})$ to the Aristotelian gravitational theory, such that we have a generalization of dipole conservation to curved and unrestricted Aristotelian geometry. Further physical implications of this theory are discussed in~\cite{Huang:2023zhp}.”

  • In page 17 in the discussion section:

“\item[Aristotelian black holes] The circularly symmetric solution described in Section \ref{sec:solutions-charges}, with negative cosmological constant, shares many properties with the BTZ black hole in General Relativity. It is therefore natural to ask whether it is possible to define a notion of an ``Aristotelian black hole’’. Given the significant differences between the properties of Aristotelian and Riemannian geometries, one might attempt to extend the concept of an event horizon to Aristotelian geometries, as well as their thermal properties, as was done, for example, in the case of Carrollian gravitational theories \cite{Ecker:2023uwm}. One possible approach is to leverage the isomorphism between the dipole algebra with a negative cosmological constant and the three-dimensional Poincaré algebra to describe thermal solutions within the Chern-Simons formulation of the Aristotelian theory. In particular, we would like to study whether the flat-space generalization of the Cardy formula \cite{Bagchi:2012xr,Barnich:2012xq} could play a significant role in describing the thermal properties of Aristotelian black holes, as well as its connection to the BMS-type asymptotic conditions outlined in \eqref{eq:as_cond}.”

---

## Editorial Decision

published